# Ballistic, maximal strength and strength-endurance performance of male handball players: Are they affected by the evaluator's sex?

Danica Janicijevic[1,2,3]*, Haijiao Mao[1], Yaodong Gu[1,2], Alejandro Pérez-Castilla[4], Jorge M. González-Hernández[5], Amador García-Ramos[4,6]

1 Research Academy of Human Biomechanics, The affiliated hospital of medical school of Ningbo University, Ningbo University, Ningbo, China, 2 Faculty of Sports Science, Ningbo University, Ningbo, China, 3 Faculty of Sport and Physical Education, The Research Centre, University of Belgrade, Belgrade, Serbia, 4 Faculty of Sport Sciences, Department of Physical Education and Sport, University of Granada, Granada, Spain, 5 Faculty of Health Science, Universidad Europea de Canarias, La Orotava, Tenerife, Spain, 6 Department of Sports Sciences and Physical Conditioning, Faculty of Education, Universidad Católica de la Santísima Concepción, Concepción, Chile

* jan.danica@gmail.com

**Data Availability Statement:** All relevant data are within the paper and its Supporting information files.

## Abstract

This study aimed to elucidate whether ballistic, maximal strength and strength-endurance performances are affected by the sex of the evaluator. Sixteen young male handball players attended two testing sessions that only differed in the sex of the evaluators (2 women *vs.* 2 men). The two sessions were performed in a counterbalanced order. Ballistic performance (countermovement jump height and throwing velocity), maximal strength performance (squat and bench press [BP] one-repetition maximum [1RM]), and strength-endurance performance (number of repetitions-to-failure in BP and average velocity of the set in the squat) were assessed in both sessions. BP 1RM was greater in the presence of women evaluators ($p = 0.036$, ES = 0.09), whereas no differences were observed for the remaining variables ($p \geq 0.254$, ES $\leq$ 0.19). Low correlations ($r$ median [range] = -0.074 [-0.693, 0.326]) were observed between the different performance tests for the percent differences between both testing sessions. The sex of the evaluators has minimal influence on a number of physical traits in young male handball players when they are tested in the presence of other members of the team, while the low correlations indicate that a higher performance in one test under the presence of women does not imply a higher performance under the presence of women in other performance tests.

## Introduction

Successful performance in many sports depends on the athlete's ability to jump, throw, kick, and run at very fast velocities for a prolonged period of time [1]. Previous studies have evidenced a strong link between the outcomes of upper- and lower- body tests and athletic

**Funding:** The author(s) received no specific funding for this work.

**Competing interests:** The authors have declared that no competing interests exist.

**Abbreviations:** 1RM, one-repetition maximum; BP, bench press; CMJ, countermovement jump; ES, Cohen's *d* effect size; *r*, Pearson's correlation coefficients.

performance [2–6]. For example, ballistic performance (e.g., jump height or throwing velocity) has been shown to be able to discriminate between athletes of different competitive levels [7, 8]. Similarly, maximal strength test outcomes, commonly assessed as the one-repetition maximum (1RM), are positively associated with sprint performance [6] and with the ranking of wrestlers [4] and judokas [5]. Besides the importance of ballistic performance and maximal strength, the ability to repeatedly develop a high level of force without decreasing performance (i.e., strength-endurance) is of paramount importance in many sports [9, 10]. For example, handball players need to perform high-intensity actions (e.g., jumps, throws, and changes of direction) with maximal performance from the beginning to the end of the match. Therefore, it is not surprising that these important and distinctive qualities (ballistic, maximal strength and strength-endurance) are frequently assessed as a part of test batteries to detect athletes' strengths and weaknesses in order to prescribe individualized training programs [10].

It is well documented that sports performance does not only depend on physiological, motor, technical, and tactical abilities, but also on social influences as it has been documented by Triplit [11] who argued that the presence of others increases arousal of the "competitive instinct" and serves to increase physical competence. This phenomenon, named "social facilitation", indicates that people perform simple tasks with a greater performance in presence of others in comparison to when they are alone [12], which has been confirmed by a number of studies. For example, Baker et al. [13] and Rhea et al. [14] demonstrated that athletes lifted more weight during a 1RM test when observers (i.e., passive spectators) were present, while Murray et al. [15] found a greater rowing performance (higher distance and power output in a 9 min self-paced rowing test conducted on an ergometer) in the presence of others even in a virtual reality environment. Furthermore, it has been noted that the performance of athletes is greater when the sport activity is performed in a group that is performing the same activity [16], in the presence of sports adversaries [14] or in the presence of spotters (i.e., a person who watches and helps someone who is performing some exercise in order to prevent potential injuries) [17]. However, one factor that has been frequently overlooked is the sex of the observer.

Although the presence of an observer of the opposite sex could affect the outcomes of physical tests, a low number of studies have analyzed this topic [18–20]. Nevertheless, those studies are generally in consent that the presence of female observers when males are evaluated produces an increase in strength performance [13] and a decrease in the ratings of perceived exertion [21]. However, when females took the role of spotters the improvements in the BP 1RM did not reach statistical significance compared to the use of male spotters [20]. Although it has been highlighted as a significant factor in other research areas (i.e., psychology, sociology, etc.) [22], there is paucity of studies exploring whether the sex of the evaluator influences athletic performance. To our knowledge, there are only two studies on this topic and both of them suggested that grip strength [23] and 50 yards dash, shuttle run test or sit ups [24] of undergraduate students were unaffected by the sex of the evaluators. However, an important aspect of these studies is that the group of subjects tested was composed by both men and women. Also, it has not been elucidated whether a possible effect of the sex of the evaluator could differ among different physical tests. Therefore, it is of interest to explore whether the evaluator's sex influences physical performance of male subjects and whether the potential effect of the evaluator's sex affects differently ballistic, maximal strength and strength-endurance performances.

To fulfil these research gaps, we designed a study with the aim to explore whether the sex of the evaluator influences ballistic (countermovement jump [CMJ] height and throwing velocity), maximal dynamic strength (squat and bench press [BP] 1RM) and strength-endurance (squat and BP sets performed against the 85% of 1RM) performance of young male handball

players. Based on the results of similar studies [13, 22], we hypothesized a higher performance for all tests under the presence of female evaluators compared to male evaluators.

## Materials and methods

### Subjects

Sixteen male handball players (age = 24.3 ± 3.5 years, body mass = 81.1 ± 14.0 kg, body height 1.76 ± 0.05 m) experienced with resistance training (2.4 ± 2.2 years) volunteered to participate in the present study. Subjects were practicing handball at least 5 years at a recreational level and were free from musculoskeletal injuries that could compromise the results of the present study. They were instructed to restrain from performing extenuating exercises over the course of the study. Subjects were blind to the study purpose and their coach only told them that the testing sessions were important to know their physical performance. All subjects signed a written consent prior to the initiation of the study. The individual in this manuscript has given written informed consent (as outlined in PLOS consent form) to publish these case details. The experiment was approved by the local Ethics Committee of the University of Granada (491/CEIH/2018) according to the Declaration of Helsinki.

### Study design

A cross-sectional study was designed to explore whether the sex of the evaluators influence ballistic, maximal dynamic strength, and strength-endurance performance in male handball players. The study consisted of two identical sessions that were separated by 48–72 hours. The only difference was the presence of either men or women as evaluators. Specifically, in one session 2 women were in charge of administering the tests and in another session 2 men were in charge of the tests. The two sessions were performed in a counterbalanced order. Both women and men evaluators were young (i.e., between 23 and 30 years) and physically active. The two female evaluators were of Chilean nationality, while the male evaluators were of Spanish nationality (See Fig 1 for testing set-up). The participants did not know any of the evaluators prior to the study. Both testing sessions were performed at the university research laboratory and at a consistent time of the day for each subject (± 1 hour).

### Testing procedures

Prior to the beginning of the study, subjects' body mass (TBF-171 300A, Tanita Corporation of America Inc., Arlington Heights, IL, USA) and body height (Seca 202, Seca Ltd., Hamburg, Germany) were measured. Both testing sessions started with the same standardized warm-up that consisted of 5 minutes of jogging at a self-selected pace, 10 push-ups, and dynamic stretching. Following the warm-up, ballistic, maximal dynamic strength and strength-endurance performance were assessed in a sequential order following the procedures described below.

**Ballistic performance.** Countermovement jump (CMJ) height: the CMJ was performed using a self-selected depth and holding the hands on the hips. Following the 3 submaximal CMJs that were used as a part of the specific warm-up, the height of 5 CMJs were recorded using a validated mobile application (MyJump2) that recorded the video-image at 240 frames per second through an iPhone 8 plus (iPhone; Apple, Inc., Cupertino, CA, USA) [25]. The rest between consecutive CMJs was 1 minute. Subjects were instructed to jump as high as possible and the average value of the 4 jumps with greater jump height was used for statistical analyses. Jump height feedback was provided by the evaluators immediately after each attempt.

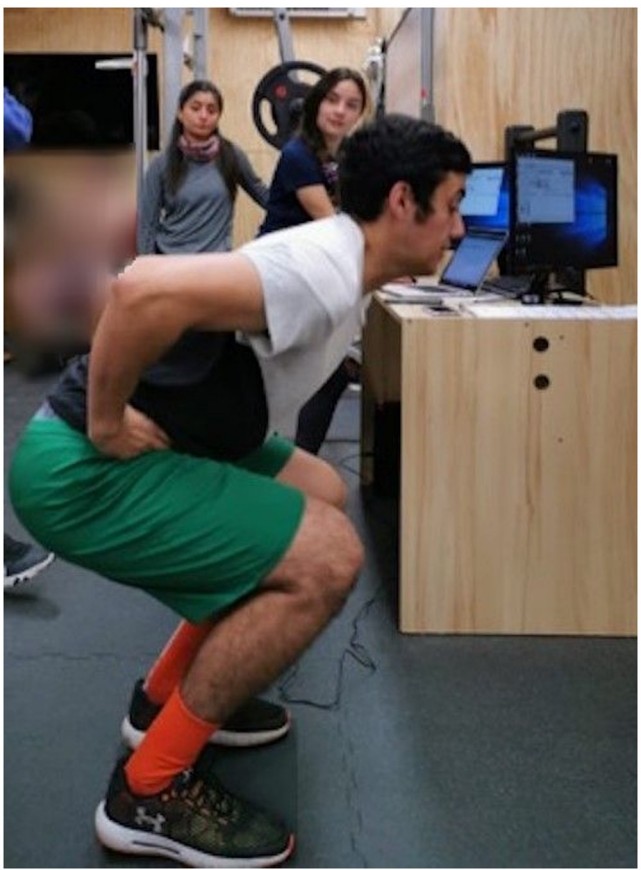

**Fig 1. A participant being tested in the countermovement jump exercise by two female evaluators.**

- Throwing velocity: following a specific warm-up (i.e., 3 submaximal handball throws), subjects performed 5 maximal handball throws with the dominant hand separated by 30 seconds of passive rest. A standing shot was performed following handball official regulations for the penalty shot (rule 14) [26]. An official handball ball size III was thrown towards the radar device (Stalker ATS II, Applied Concepts, Dallas, TX, USA) which was hold by the evaluator. The evaluator was standing behind a secure stopping net at 5 m of distance from the subjects. Throwing velocity feedback (km·h$^{-1}$) was provided following each throw, while the average value of the 4 fastest throws was used for statistical analyses.

**Maximal dynamic strength performance.** An incrementing loading test was used to estimate the 1RM through the individualized load-velocity relationship during the squat and BP exercises performed in a Smith Machine (FFittech, Taiwan, China). Details of the procedure used to estimate the 1RM have been described elsewhere [27]. The loading test consisted of six loads (40%, 50%, 60%, 70%, 80%, and 90% of their self-reported 1RM). Subjects performed two trials with each load separated by a rest of 30 seconds for light loads (40%, 50%, and 60% of 1RM) and 1 minute for heavy loads (70%, 80%, and 90% of 1RM). The rest between repetitions with different loads was set to 3 minutes. Subjects were instructed to lift the barbell at maximal intended velocity and the mean velocity of the lifting phase was recorded with a linear velocity transducer (T-Force System, Ergotech, Murcia, Spain) that was attached to the barbell of the Smith machine. The 1RM was estimated from the individualized load-velocity

relationships as the load associated with the velocity of the 1RM (i.e., 0.30 m·s$^{-1}$ for the squat and 0.17 m·s$^{-1}$ for the BP).

**Strength-endurance performance.** Subjects performed against the 85% of 1RM one set of six repetitions in the squat exercise and one set of repetitions to failure in the BP exercise. Both tests were separated by 10 minutes. The average velocity of the six repetitions in the squat were recorded with the linear velocity transducer and the average value of the six repetitions was used for statistical analyses. The number of repetitions performed to failure was the dependent variable considered for the BP exercise. Subjects were instructed to perform the lifting phase of all repetitions as fast as possible. Subjects did not perform repetitions to failure during the squat exercise because they do not use this training strategy in this exercise during their usual training.

### Statistical analyses

Descriptive data are presented as means and standard deviations. The normality of the data was confirmed by the Shapiro-Wilk test ($p > 0.05$). Student's paired-sample t-tests and the Cohen's d effect size (ES) were used to compare the magnitude of ballistic (CMJ height and throwing velocity), maximal dynamic strength (squat and BP 1RM), and strength-endurance (number of repetitions-to-failure in BP and average velocity of the set in the squat) performances between both testing sessions that only differed in the sex of the evaluators (2 women *vs*. 2 men). Pearson's correlation coefficients ($r$) were calculated to determine whether the percent differences in the magnitude of the variables obtained in the two testing sessions were associated between the different tests. The $r$ coefficients were also used to explore the association of the six dependent variables between both testing sessions. The magnitude of the ES was interpreted as follows: trivial ($< 0.20$), small (0.20–0.59), moderate (0.60–1.19), large (1.20–2.00), and very large ($> 2.00$) [28]. Qualitative interpretations of the $r$ coefficients as defined by Hopkins et al. [28] (0.00–0.09 trivial; 0.10–0.29 small; 0.30–0.49 moderate; 0.50–0.69 large; 0.70–0.89 very large; 0.90–0.99 nearly perfect; 1.00 perfect) are also provided. All statistical analyses were performed using SPSS software version 20.0 (IBM SPSS Inc., Chicago, IL, USA) and statistical significance was set at an alpha level of 0.05.

### Results

Descriptive data of the performance variables measured in both testing sessions are depicted in Table 1. The BP 1RM was the only variable that showed to be greater in the presence of women evaluators ($p = 0.036$, ES = 0.09), while the magnitudes of the other performance variables were unaffected by the sex of the evaluator ($p \geq 0.254$, ES $\leq 0.19$) (Fig 2).

Low correlations ($r$ median [range] = -0.074 [-0.693, 0.326]) were generally observed between the different tests for the percent differences in performance observed between both

**Table 1. Descriptive data of the dependent variables.**

| | Ballistic performance | | Maximal strength | | Strength-endurance | |
|---|---|---|---|---|---|---|
| | CMJ height (cm) | Throwing velocity (km·h$^{-1}$) | Squat 1RM (kg) | BP 1RM (kg) | Squat set velocity (m·s$^{-1}$) | BP repetitions to failure (u.a.) |
| Women evaluators | 32.0 ± 5.6 | 71.7 ± 7.4 | 132.9 ± 27.2 | 81.1 ± 21.6 | 0.51 ± 0.07 | 9.4 ± 2.4 |
| Men evaluators | 33.0 ± 4.9 | 71.3 ± 7.9 | 131.3 ± 27.4 | 79.2 ± 21.3 | 0.51 ± 0.06 | 8.8 ± 3.3 |
| $r$ | 0.87 | 0.91 | 0.88 | 0.99 | 0.54 | 0.82 |

CMJ, countermovement jump; BP, bench press; 1RM, one-repetition maximum. Data are presented as mean ± standard deviation. $r$, association of the dependent variables between both testing sessions.

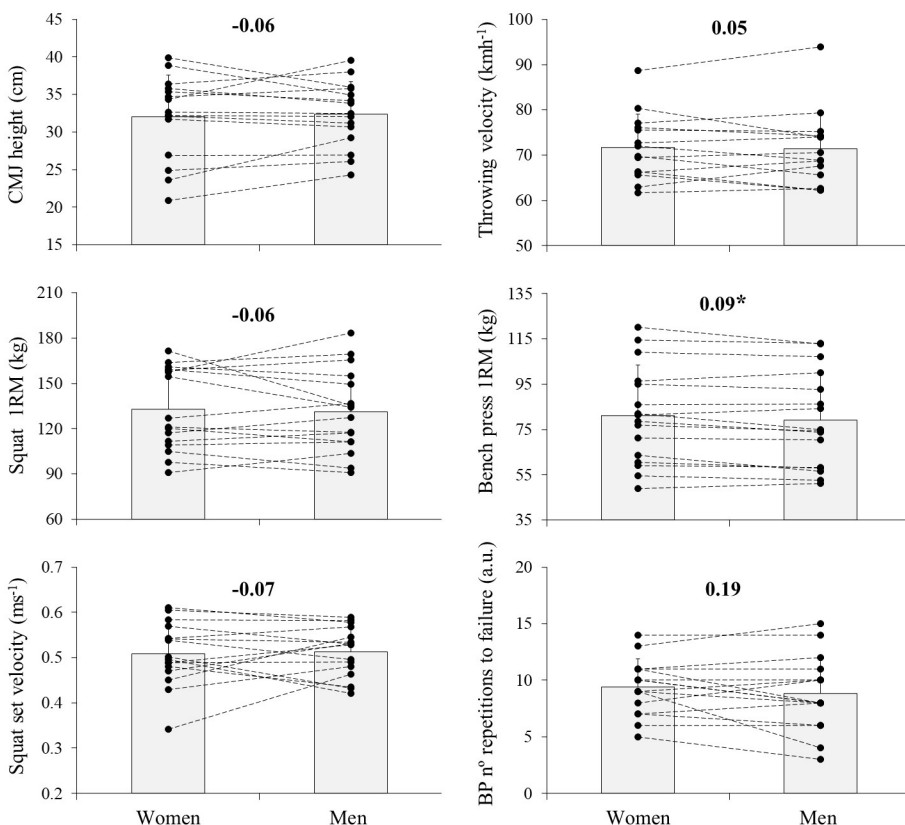

**Fig 2. Comparisons of ballistic (countermovement [CMJ] height and throwing velocity), maximal dynamic strength (squat and BP one-repetition maximum [1RM]) and strength-endurance (number of repetitions-to-failure in BP and average velocity of the set in the squat) performance between both testing sessions that differed in the sex of the evaluators.** The Cohen's d effect size ([women evaluators—men evaluators] / SD both; bold numbers) are indicated. Statistical significance: * P < 0.05. Data are presented as means and standard deviations.

sessions, indicating that a higher performance in one test under the presence of women does not imply a higher performance under the presence of women in other performance tests (Table 2). In fact, the only significant correlation (p ≤ 0.01) was an inverse association (r = -0.693) between the BP 1RM and the average velocity of the six repetitions performed during the squat exercise.

**Table 2. Associations between the different tests for the percent differences in performance observed between both testing sessions.**

|  | CMJ height | Throwing velocity | Squat 1RM | BP 1RM | Squat set velocity |
|---|---|---|---|---|---|
| Throwing velocity | 0.230 |  |  |  |  |
| Squat 1RM | 0.063 | -0.443 |  |  |  |
| BP 1RM | 0.204 | -0.108 | -0.001 |  |  |
| Squat set velocity | -0.149 | -0.099 | 0.270 | **-0.693** |  |
| BP repetitions to failure | -0.344 | -0.386 | 0.060 | 0.326 | -0.074 |

CMJ, countermovement jump; BP, bench press; 1RM, one-repetition maximum. Data are presented as Pearson's correlation coefficients (r). Bold number indicates a significant correlation (p ≤ 0.01).

## Discussion

This study was designed to explore whether the sex of the evaluators influence ballistic (CMJ height and throwing velocity), maximal dynamic strength (squat and BP 1RM) and strength-endurance (squats and BP sets performed against the 85% of 1RM) performance of young male handball players. The main findings of this study were that (1) only the BP 1RM was significantly greater under the presence of women evaluators, but no significant differences were observed for the remaining variables, and (2) low correlations were observed between the different performance tests for the percent differences in performance observed between both testing sessions. These findings suggest that the sex of the evaluator has minimal influence on a number of physical traits in young recreationally active male handball players and that subjects who showed a higher performance under the presence of women evaluators in one test do not necessarily show a higher performance under the presence of women evaluators in other tests.

Ballistic, maximal dynamic strength and strength-endurance performance were unaffected by the sex of the evaluator, with the only exception of the BP 1RM which was greater in the presence of women evaluators. The greater BP 1RM observed in the present study is in line with the findings of Baker et al. [13] who also reported a greater BP 1RM in the presence of women spectators. It is not clear, however, why the performance of the BP 1RM was the only test that resulted to be influenced by the presence of women evaluators, while the squat 1RM, or ballistic and strength-endurance tests were not affected. The only two studies that explored the influence of the sex of the evaluators on the outcomes of physical tests reported that neither the maximal strength (i.e., hand grip) [23], nor the 50 yards dash, shuttle run test or sit ups [24] were affected by the sex of the evaluator. A possible explanation for the studies of Rikli [23, 24] could be the fact that their study samples were composed by both males and females. However, we observed similar results although our sample consisted of exclusively male subjects. To our knowledge, our study was the first to examine the influence of the sex of the evaluator on ballistic and strength-endurance performance. The contradicting findings of our study with respect to previous research [13] might emerge from the fact that we included women as evaluators, while other authors designed a study in which women had a more passive role (i.e., spectators). These findings highlight that more studies are needed to obtain stronger conclusions about this topic.

The sex of the evaluator did not have a consistent effect on individual subjects for the performance in different physical tests. This is evidenced by the generally low correlations (r median = -0.074) observed between the different tests for the percent differences in performance measured between both sessions. This finding is in line with the only study that has explored the differences in the percent increments in strength performance obtained in the presence of women and men. Specifically, Baker et al. [19] did not report significant differences in the BP 1RM (p = 0.08) nor in the leg press 1RM (p = 0.71) between sessions that differed in the sex of the spectator. In the present study we only observed a significant, but negative, correlation between the differences in the outcomes for the BP 1RM (maximal strength test) and the velocity of the squat set (strength-endurance test). The negative correlation indicates that subjects who performed the BP 1RM test better in the presence of women evaluators tended to have a lower velocity in the squat set in the presence of women evaluators. Given that in the present study only 1 out of 15 correlations reached statistical significance, it is likely that the significant correlation may be anecdotal and promoted by a type I error.

Although our study provides novel information regarding the effect of the sex of the evaluator on a number of physical traits, several limitations should be acknowledged when interpreting the findings of our study. Firstly, we did not include group of women subjects, which

would enable a deeper assessment of effect that evaluators of different sex might have on the sports performance of female athletes. Secondly, we did not take into account the influence that the presence of other subjects (other members of the team) might have on the performance of the tests. It is possible that a potential positive effect caused by women evaluators on strength performance was counteracted because other players were present as observers during the test and this could also increase the performance of the players [13, 14]. Therefore, future studies should try to explore the influence of men and women evaluators on sports performance of the men and women subjects without the presence of observers (e.g., players from the same team). Finally, it is plausible that the results of this study would differ with a different set of male and female evaluators, being needed more studies addressing similar research questions to reach more generalizable conclusions.

In conclusion, ballistic, maximal dynamic strength and strength-endurance performance of young male handball players seem to be trivially affected by the sex of the evaluators. The BP 1RM was the only variable that showed a significantly higher performance under the presence of women evaluators, but the gains were marginal (ES = 0.09). In addition, subjects that showed a higher performance under the presence of women evaluators in one test do not necessarily present a higher performance in other tests (no significant correlations were observed). The results of the present study collectively suggest that the sex of the evaluator is not an important factor to consider when the strength performance of young handball players is tested in the presence of other members of the team. Therefore, when physical tests are implemented in the presence of other members of the team, which is the most common procedure in practice, male handball players can be indistinctly assessed by male or female evaluators. Future studies should be implemented to explore the effect of the sex of the evaluators when athletes are tested without the presence of observers (e.g., other team members).

## Supporting information

**S1 Data.**
(XLSM)

## Acknowledgments

This work was performed under the projects 451-03-68/2020-14/200015 and 451-03-68/2020-14/200021 assigned from the Ministry of education, science and technological development of Republic of Serbia.

## Author Contributions

**Conceptualization:** Alejandro Pérez-Castilla, Jorge M. González-Hernández, Amador García-Ramos.

**Data curation:** Danica Janicijevic, Alejandro Pérez-Castilla, Amador García-Ramos.

**Investigation:** Danica Janicijevic, Alejandro Pérez-Castilla.

**Methodology:** Danica Janicijevic, Haijiao Mao.

**Resources:** Yaodong Gu.

**Supervision:** Yaodong Gu, Amador García-Ramos.

**Writing – original draft:** Danica Janicijevic.

**Writing – review & editing:** Danica Janicijevic, Haijiao Mao, Yaodong Gu, Alejandro Pérez-Castilla, Jorge M. González-Hernández, Amador García-Ramos.

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
