## [Decision Letter · Decision Letter 0]

15 Feb 2021

PONE-D-20-34628

Ballistic, maximal strength and strength-endurance performance of male handball players: are they affected by the evaluator's sex?

PLOS ONE

Dear Dr. Janicijevic,

Thank you for submitting your manuscript to PLOS ONE. After careful consideration, we feel that it has merit but does not fully meet PLOS ONE’s publication criteria as it currently stands. Therefore, we invite you to submit a revised version of the manuscript that addresses the points raised during the review process.

All reviewers found merit in your manuscript. However, all three indicated that there needed to be much greater detail regarding the methodology used before they would be able to fully evaluate your manuscript. Please carefully consider each of their points prior to resubmission. Of note, Reviewer 3 uploaded comments as an attachment.

We look forward to receiving your revised manuscript.

Kind regards,

Jeremy P Loenneke

Academic Editor

PLOS ONE

Journal Requirements:

2.Thank you for stating the following in the Acknowledgments Section of your manuscript:

"This work was supported by the grants 175037 and 175012 from the Ministry of education, science and technological development of Republic of Serbia."

Reviewers' comments:

Reviewer's Responses to Questions

**Comments to the Author**

1. Is the manuscript technically sound, and do the data support the conclusions?

Reviewer #1: No

Reviewer #2: Yes

Reviewer #3: Partly

2. Has the statistical analysis been performed appropriately and rigorously? 

Reviewer #1: Yes

Reviewer #2: Yes

Reviewer #3: No

3. Have the authors made all data underlying the findings in their manuscript fully available?

Reviewer #1: Yes

Reviewer #2: Yes

Reviewer #3: Yes

4. Is the manuscript presented in an intelligible fashion and written in standard English?

Reviewer #1: Yes

Reviewer #2: Yes

Reviewer #3: Yes

5. Review Comments to the Author

Reviewer #1: The authors examine how the sex of the test administrator impact various performance measures among male handball players.

The primary concern I have with the present study relates to how the two male test administrators and two female test administrators are representative of all male and female test administrators. How we can be sure that these results would hold true with a different set of male and female administrators? I find it difficult to conclude that the results of this study are generalizable to all male and female administrators. How would a different demographic of male and female administrators change the results of this study? This should at the very least be mentioned as a limitation of the study, but in my opinion, this represents a fatal flaw. At the very least more information should be included about the test administrators including age and demographics.

Was the order of male and female administrators counterbalanced, or did all participants always have the same sex administer the first session? This should be added and would be a limitation if the same sex always administered the first session.

What were the participants told that the purpose of the study was? Were they aware that you were testing how their performance would differ when either males or females were administering the tests? Were they made aware of the results of previous research? This should be mentioned as it may have biased the results.

“The 1RM was estimated from the individualized load-velocity relationships as the load associated with the velocity of the 1RM (i.e., 0.30 m·s-1 for the squat and 0.17 m·s-1 for the BP).” Why not just test the 1RM?

What is the reliability of all these measurements? Based on the sample size, the reliability of these measurements would have to be very high to be sure the study was adequately powered.

Reviewer #2: Thank you for allowing me to review this manuscript. I actually find it very interesting. However, there are some issues that need addressing in order to clarify a few things. For example, the whole emphasis on the paper is the impact of male and female evaluators. Yet, there is very little detail regarding their roles in the study methodology. Please be more specific in the methods section regarding this matter. Moreover, there is no discussion about whether males were selected as evaluators for the 1st visit, 2nd visit, or whether evaluator sex was randomized across visits. Without knowing this, it is possible that your results are biased. Therefore, the authors need to take these things into consideration and improve the methods section. Specific details regarding the manuscript are provided below:

Introduction

Line 58: Please list the type of performances for each sport that 1RM is associated with.

Line 61: List some examples of why muscular endurance is important.

Line 65-66: Needs to be fixed. It is an incomplete sentence in its current status.

Line 67-69: what does liberate latent energy translate to in terms of performance? In other words, what performance test were evaluated for this particular reference?

Line 74: Please be more specific about the type of rowing performance evaluation that was completed.

Line 81: you state a limited number of studies (plural), but only list one reference. Other than reference 18, what other researchers have evaluated this topic?

Line 101 and 102: Need to provide references to support your hypothesis statement.

Methods

Line 119-121: Did any of the subjects ask why evaluators were different between session 1 and 2? If yes, what did you tell them?

Line 137: What is meant by 3 probe CMJ?

Line 141: Does this mean that 4 out of the 5 jump heights were used for analysis? If so why select 4 jumps to analyze instead of peak?

Line 142: This is really important to discuss in greater detail. What kind of feedback was provided? How do you know that feedback was the same between male and female evaluators? What if males provided more encouraging feedback than females or vice versa?

Line 151: Similar to CMJ, why analyze 4 out of 5? That seems subjective.

Line 157: Although used in previous research, six warm-up loads is too excessive. Would have been better to follow NSCA guidelines and have less warm-up loads.

Line 167: To clarify, did the authors predict 1RM via velocity instead of obtaining a measured 1RM? If yes, why was that done if subjects were near their max 1RM (90% 1RM) when completing the warm-up loads? Also, how are authors going about selecting the expected 1RM velocity to develop the load velocity profiles, if 1RM was not measured? It says 0.30 and 0.17 m/s, but how were these numbers determined if 1RM was not measured? Please clarify.

Line 168: Why 6RM for squat and RTF for bench press?

Results

Tables should be made on a separate document.

Line 203-209: Remove the figure caption after the references. It doesn’t belong here.

Figure 1: I would remove this figure. It doesn’t seem necessary.

Reviewer #3: This manuscript is well-written and may be of broad applicability to those interested in exercise science, individual sport performance, and social impacts on group dynamics. The authors may wish to consider correcting for multiple comparisons, as well as offering additional justification for the study design.

6. PLOS authors have the option to publish the peer review history of their article (what does this mean?). If published, this will include your full peer review and any attached files.

Reviewer #1: No

Reviewer #2: No

Reviewer #3: No

---

## [Author Response · Author response to Decision Letter 0]

5 Mar 2021

Dear Editor,

Please find enclosed a revision of our manuscript, “Ballistic, maximal strength and strength-endurance performance of male handball players: are they affected by the evaluator's sex?” We would like to thank you for giving us the opportunity to revise and improve our manuscript, and the reviewers for their thoughtful and constructive comments. Changes to the original manuscript are highlighted in the track changes mode of Microsoft Word. An itemized point-by-point response to the reviewers’ comments is presented below.

AUTHORS’ RESPONSE TO THE REVIEWERS

Reviewer 1 

GENERAL COMMENTS

The authors examine how the sex of the test administrator impact various performance measures among male handball players. The primary concern I have with the present study relates to how the two male test administrators and two female test administrators are representative of all male and female test administrators. How we can be sure that these results would hold true with a different set of male and female administrators? I find it difficult to conclude that the results of this study are generalizable to all male and female administrators. How would a different demographic of male and female administrators change the results of this study? This should at the very least be mentioned as a limitation of the study, but in my opinion, this represents a fatal flaw. At the very least more information should be included about the test administrators including age and demographics.

RESPONSE

We understand the reviewer´s concern. We have added more information about the test administrators into the methods section: “The two sessions were performed in a counterbalanced order. Both women and men evaluators were young (i.e., between 23 and 30 years) and physically active. The two female evaluators were of Chilean nationality, while the male evaluators were of Spanish nationality. The participants did not know any of the evaluators prior to the study.” We have also added a figure depicting the two female evaluators during the CMJ test. In addition, the following information was added to the limitations section: “Finally, it is plausible that the results of this study would differ with a different set of male and female evaluators, being needed more studies addressing similar research questions to reach more generalizable conclusions”. 

We agree with the reviewer that the results of this study are not generalizable across all male and female administrators, and even with the same test administrator the results could differ if other populations are tested. However, this article adds evidence about the effect of the gender of the evaluator on males´ physical performance, but as most of the papers in the scientific literature, we do not provide a universal and definitive answer to this research question. However, we provide experimental data in a population of active athletes that contribute to expand our knowledge about this topic. 

SPECIFIC COMMENTS

COMMENT

Was the order of male and female administrators counterbalanced, or did all participants always have the same sex administer the first session? This should be added and would be a limitation if the same sex always administered the first session. 

RESPONSE

The order of the two sessions was counterbalanced between subjects. This information has been specified both in the abstract and main text in the revised version of the manuscript (in Study design section). 

COMMENT

What were the participants told that the purpose of the study was? Were they aware that you were testing how their performance would differ when either males or females were administering the tests? Were they made aware of the results of previous research? This should be mentioned as it may have biased the results.

RESPONSE

This information has been clarified in the revised version of the manuscript and now it reads: “Subjects were blind to the study purpose and their coach only told them that the testing sessions were important to know their physical performance” (in Subjects section).

COMMENT

“The 1RM was estimated from the individualized load-velocity relationships as the load associated with the velocity of the 1RM (i.e., 0.30 m·s-1 for the squat and 0.17 m·s-1 for the BP).” Why not just test the 1RM?

RESPONSE

This is a pertinent question. We used this approach because (I) the assessed handball players do not usually perform 1RM lifts during their training programs, and (II) the available literature indicate that the individualised load-velocity relationship can be used to accurately estimate the 1RM (Weakley et al., Velocity-Based Training: From Theory to Application; Strength and Conditioning Journal; doi: 10.1519/SSC.0000000000000560).

COMMENT

What is the reliability of all these measurements? Based on the sample size, the reliability of these measurements would have to be very high to be sure the study was adequately powered.

RESPONSE

This is another pertinent comment. We did not calculate the reliability of the variables in our study because the two sessions were conducted in the presence of different evaluators. Anyway, the variables used in this study (CMJ height, throwing velocity, 1RM estimated through individual load-velocity profiles, lifting velocity, and repetitions to failure) are standard tests that have shown an acceptable reliability and are very commonly used both in practice and in scientific studies. Therefore, we believe that the reliability of the variables is not an issue in the present study. In addition, we have calculated the r coefficients between both testing sessions for the six dependent variables and this information was added to the Table 1 in the revised version of the manuscript (r ranged from 0.99 to 0.54). Note also that in addition to p-values, we reported and discussed the magnitude of the effect sizes. We would like to thank the reviewer for the positive feedback that have contributed to improve the quality of the information presented in our original submission.

Reviewer #2:

GENERAL COMMENT

Thank you for allowing me to review this manuscript. I actually find it very interesting. However, there are some issues that need addressing in order to clarify a few things. For example, the whole emphasis on the paper is the impact of male and female evaluators. Yet, there is very little detail regarding their roles in the study methodology. Please be more specific in the methods section regarding this matter. Moreover, there is no discussion about whether males were selected as evaluators for the 1st visit, 2nd visit, or whether evaluator sex was randomized across visits. Without knowing this, it is possible that your results are biased. Therefore, the authors need to take these things into consideration and improve the methods section. Specific details regarding the manuscript are provided below:

RESPONSE

We would like to thank the reviewer for the positive feedback that have contributed to improve the quality of our original submission. We have specified in the revised version of the manuscript that “The two sessions were performed in a counterbalanced order”. We have also provided more detail regarding the roles of male and female evaluators: “The only difference was the presence of either men or women as evaluators. Specifically, in one session 2 women were in charge of administering the tests and in another session 2 men were in charge of the tests. The two sessions were performed in a counterbalanced order. Both women and men evaluators were young (i.e., between 23 and 30 years) and physically active. The two female evaluators were of Chilean nationality, while the male evaluators were of Spanish nationality. The participants did not know any of the evaluators prior to the study”.

SPECIFIC COMMENTS

COMMENT

Line 58: Please list the type of performances for each sport that 1RM is associated with.

RESPONSE

This sentence has been modified and now it reads: “Similarly, maximal strength test outcomes, commonly assessed as the one-repetition maximum (1RM), are positively associated with sprint performance (Comfort et al., 2014) and with the ranking of wrestlers (García-Pallarés et al., 2011) and judokas (Franchini et al., 2011).”

COMMENT

Line 61: List some examples of why muscular endurance is important.

RESPONSE

This sentence was added to the revised version of the manuscript: “For example, handball players need to perform high-intensity actions (e.g., jumps, throws, and changes of direction) with maximal performance from the beginning to the end of the match.”

COMMENT

Line 65-66: Needs to be fixed. It is an incomplete sentence in its current status.

Line 67-69: what does liberate latent energy translate to in terms of performance? In other words, what performance test were evaluated for this particular reference?

RESPONSE

These sentences have been modified and now it reads: “It is well documented that sports performance does not only depend on physiological, motor, technical, and tactical abilities, but also on social influences as it has been documented by Triplit [11] who argued that the presence of others increases arousal of the “competitive instinct” and serves to increase physical competence.”

COMMENT

Line 74: Please be more specific about the type of rowing performance evaluation that was completed.

RESPONSE

This sentence has been modified and now it reads: “Murray et al. [15] found a greater rowing performance (higher distance and power output in a 9 min self-paced rowing test conducted on an ergometer) in the presence of others even in a virtual reality environment.”

COMMENT

Line 81: you state a limited number of studies (plural), but only list one reference. Other than reference 18, what other researchers have evaluated this topic? 

RESPONSE

Thank you for this appreciation. We have added two more articles. 

COMMENT

Line 101 and 102: Need to provide references to support your hypothesis statement.

RESPONSE

This sentence has been rewritten and now it reads: “Based on the results of similar studies [13,22], we hypothesized a higher performance for all tests under the presence of female evaluators compared to male evaluators”.

COMMENT

Line 119-121: Did any of the subjects ask why evaluators were different between session 1 and 2? If yes, what did you tell them?

RESPONSE

Some subjects made some comments that they were happier with women evaluators, but they were blinded to the study purpose as we have indicated in the revised version of the manuscript. None of them asked the reason of different evaluators in both testing sessions. 

COMMENT

Line 137: What is meant by 3 probe CMJ?

RESPONSE

We have modified this sentence and now it reads: “Following the 3 submaximal CMJs that were used as a part of the specific warm-up, the height of 5 CMJs…”.

COMMENT

Line 141: Does this mean that 4 out of the 5 jump heights were used for analysis? If so why select 4 jumps to analyze instead of peak?

RESPONSE

This is a pertinent question. We used the average value instead of the peak based on a meta-analysis conducted by Claudino et al. (J Sci Med Sport. 2017 Apr;20(4):397-402. doi: 10.1016/j.jsams.2016.08.011) who concluded that “The average CMJ height was more sensitive than highest CMJ height in monitoring neuromuscular status.” We decided to delete the worst jump in case that subjects for any reason were not fully activated in a particular trial. Anyway, the main findings of the present study are not expected to be affected by the procedure used to select the final test outcome. 

COMMENT

Line 142: This is really important to discuss in greater detail. What kind of feedback was provided? How do you know that feedback was the same between male and female evaluators? What if males provided more encouraging feedback than females or vice versa?

RESPONSE

The only feedback that was provided was the jump height or the throwing velocity. Before each trial subjects were told that they needed to jump as high as possible or throw the ball as fast as possible, and after each trial they were informed about their performance in the test (how much they jump or which velocity the ball achieved). Same and very simple information was given by the evaluators so we do not expect that this could affect the outcomes of the present study due to the simplicity of the information provided by the evaluators. 

COMMENT

Line 151: Similar to CMJ, why analyze 4 out of 5? That seems subjective.

RESPONSE

Same reason as indicated above. We agree that the procedure is subjective and we used the procedure that we think it could provide more reproducible outcomes. However, please note that there is no consensus in the scientific literature about which procedure should be used to select the final test outcome. In fact, when the variables are obtained with a high reliability (e.g., CMJ height and throwing velocity), it does not really matter if the peak or the average performance is used for the comparisons. This was recently confirmed by study conducted by our research group exploring the throwing velocity: 

- Chirosa et al. Reliability of Throwing Velocity during Non-specific and Specific Handball Throwing Tests. Int J Sports Med. 2020 Oct 30. doi: 10.1055/a-1273-8630. Online ahead of print.

COMMENT

Line 157: Although used in previous research, six warm-up loads is too excessive. Would have been better to follow NSCA guidelines and have less warm-up loads.

Line 167: To clarify, did the authors predict 1RM via velocity instead of obtaining a measured 1RM? If yes, why was that done if subjects were near their max 1RM (90% 1RM) when completing the warm-up loads? Also, how are authors going about selecting the expected 1RM velocity to develop the load velocity profiles, if 1RM was not measured? It says 0.30 and 0.17 m/s, but how were these numbers determined if 1RM was not measured? Please clarify.

RESPONSE

The available literature indicates that the load-velocity relationship can be used to accurately estimate the 1RM (Weakley et al., Velocity-Based Training: From Theory to Application; Strength and Conditioning Journal; doi: 10.1519/SSC.0000000000000560). We used the general velocity of the 1RM reported in the systematic review mentioned above. Please, note that the velocity of the individual 1RM is not a reproducible metric, and that all studies conducted to date that have compared the precision of a general or individual V1RM to estimate the 1RM through the individualised load-velocity relationship have not find significant differences between both V1RMs. We agree with the reviewer that less loads could be implemented to determine the load-velocity profile, but to avoid potential problems with reviewers, we decided to test more loads. But yes, likely by testing only the 50% and 90% of 1RM we would have obtained the same results as we have observed in previous studies conducted by our research group (for example, see: García-Ramos et al., Int J Sports Physiol Perform. 2018 Apr 1;13(4):474-481. doi: 10.1123/ijspp.2017-0374.). Handball players do not usually perform 1RM lifts during their training programs and they did not want to perform 1RM trials for the purpose of this study. Anyway, by lifting the 90%1RM load we are sure to have an experimental point very close to the 1RM ensuring an accurate prediction. The high precision could be further supported by the high association observed for the 1RM during the squat (r = 0.88) and bench press (r = 0.99) that we have reported in the revised version of Table 1.

COMMENT

Line 168: Why 6RM for squat and RTF for bench press?

RESPONSE

This is a pertinent question. This decision is justified because handball players were not accustomed to performing sets to failure during the squat, but they were familiar with repetitions to failure during the bench press. The following information was added to the revised version of the manuscript: “Subjects did not perform repetitions to failure during the squat exercise because they do not use this training strategy in this exercise during their usual training”.

Results

COMMENT

Tables should be made on a separate document.

RESPONSE

We appreciate the reviewer´s comment but the instruction for authors of PlosOne indicate the following: “Tables are inserted immediately after the first paragraph in which they are cited”.

COMMENT

Line 203-209: Remove the figure caption after the references. It doesn’t belong here.

RESPONSE

We appreciate the reviewer´s comment but the instruction for authors of PlosOne indicate the following: “Figure captions are inserted immediately after the first paragraph in which the figure is cited. Figure files are uploaded separately.”.

COMMENT

Figure 1: I would remove this figure. It doesn’t seem necessary.

RESPONSE

We agree with the reviewer. This figure has been removed and replaced it by another figure showing the 2 female evaluators during a CMJ test. 

Reviewer #3: 

GENERAL COMMENT

This manuscript is well-written and may be of broad applicability to those interested in exercise science, individual sport performance, and social impacts on group dynamics. The authors may wish to consider correcting for multiple comparisons, as well as offering additional justification for the study design. Overall, this is a well-written manuscript that may contribute to the aims of PlosOne and to relevant scientific literature. A paucity of work has explored the impact of evaluator sex on exercise testing outcomes. This study was designed to extend work in this domain and offers novel directions for continued research. There are certain concerns that I feel should be addressed to shift this paper closer to publication readiness. 

RESPONSE

We would like to thank the reviewer for the positive and constructive feedback that have contributed to improve the quality of our original submission. Specific responses can be found below.

SPECIFIC COMMENTS

COMMENT

1. In addition to social facilitation, some research brings up the converse phenomenon of social loafing. I think it is worth highlighting this distinction, particularly as social loafing is often observed in group contexts when individuals are working collectively towards a goal, and other male handball players (of the same team affiliation) were present during testing. 

RESPONSE

We appreciate this comment, however, we think that the phenomenon social loafing is not relevant for this study since the participants were not collectively working towards a same goal, but rather their individual abilities were assessed in the presence of their team players. Since they represent competitive sports group, we think that the presence of others could just enhance their physical performance as it has been shown in previous studies (Rhea et al., 2003; Baker, Jung and Petrella, 2011; Murray et al., 2016).

Baker, S.C., Jung, A.P. and Petrella, J.K. (2011) Presence of Observers Increases One Repetition Maximum in College-age Males and Females. International journal of exercise science 4, 199–203.

Murray, E.G., Neumann, D.L., Moffitt, R.L. and Thomas, P.R. (2016) The effects of the presence of others during a rowing exercise in a virtual reality environment. Psychology of Sport and Exercise 22, 328–336.

Rhea, M.R., Landers, D.M., Alvar, B.A. and Arent, S.M. (2003) The effects of competition and the presence of an audience on weight lifting performance. Journal of Strength and Conditioning Research 17, 303–306.

COMMENT

2. It is not a flaw of the study, but a clear justification of why the impact of evaluator sex on female handball players’ performance was not evaluated is needed. 

RESPONSE

We agree with the reviewer that our manuscript would be stronger by including a group of female handball players. However, we did not have such a sample available at the time of the study. This has been highlighted in the limitations section.

COMMENT

3. Evidence of a sample size estimation should be provided to ensure that this study was adequately powered to detect an experimental effect. 

RESPONSE

This is a pertinent comment. We did not conduct a sample size estimation because we tested all possible handball players of the team. Note also that the reliability of the variables analysed in our study (CMJ height, throwing velocity, 1RM estimated through individual load-velocity profiles, lifting velocity, and repetitions to failure) are standard tests that have shown an acceptable reliability and are very commonly used both in practice and in scientific studies. Therefore, we believe that the reliability of the variables is not an issue in the present study. In addition, we have calculated the r coefficients between both testing sessions for the six dependent variables and this information was added to the Table 1 in the revised version of the manuscript (r ranged from 0.99 to 0.54). Finally, note that in addition to p-values, we reported and discussed the magnitude of the effect sizes (all ES < 0.20). Therefore, we believe that the main conclusions of the present study are not affected by a low sample size (n = 16), especially considering that it was a repeated-measures design and that the dependent variables generally present a high reliability.

COMMENT

4. On lines 115-116, I am confused by the heading “Experimental Design,” which is followed by the text “a cross-sectional study was designed.” Cross-sectional research is non-experimental because the groups are not randomly selected, and the independent variable is not manipulated. Please revise to clarify. 

RESPONSE

We agree with the reviewer´s appreciation. We have replaced “experimental design” by “study design”.

COMMENT

5. Additional information about the male and female evaluators would be helpful. 

- What were the ages of the evaluators? 

- Did the evaluators know any of the participants or vice-versa? 

- Why did you choose to have the evaluators adopt a test-administration role, rather than a passive observer, or a (potentially) supportive role as a spotter? It is important to justify this decision because the dynamic of authority could arguably have an impact on handball players’ perceptions of evaluators. You also emphasize this test-administrator/passive observer dichotomy in the discussion, so additional elaboration is warranted. 

RESPONSE

The following information was added to the methods section: “The two sessions were performed in a counterbalanced order. Both women and men evaluators were young (i.e., between 23 and 30 years) and physically active. The two female evaluators were of Chilean nationality, while the male evaluators were of Spanish nationality. The participants did not know any of the evaluators prior to the study.” We have also added a figure depicting the two female evaluators during the CMJ test. As we have noted in the manuscript, previous studies have examined the effect of the sex of the evaluator on physical performance when they act as observers or spotters, so we decided to expand the information exploring their effect when they act as an administrator of the tests. We believe that all roles are equally important to investigate, and we decided to use the role of the administrator of the tests because it has received less scientific attention. 

COMMENT

6. One concern is that there was no inclusion of a control group. That is, we cannot confirm that the presence of female evaluators is meaningfully associated with male handball player BP performance, relative to the presence of male evaluators. It could be that unmeasured variables impacted study outcomes. A control group with no evaluators would be the most ideal to establish a baseline index of performance. However, if participant safety or protocol adherence is a concern, such that participants cannot be tested alone, then another approach may be to employ mixed-sex evaluation in a separate condition. If significant effects were to emerge with same sex evaluation compared to simultaneous evaluations conducted by one male and one female evaluator, then this evidence may be more convincing. 

RESPONSE

We agree that a control group with no evaluators would be the most ideal to establish a baseline index of performance, but this is not a real scenario because athletes do not typically auto-administer their physical tests. We decided to implement testing in a real scenario where team sport athletes are tested in the presence of other players, and we only manipulated the sex of the evaluators (2 women vs. 2 men). We agree that many different combinations are valuable to address different related research questions, but in this study we were interested in elucidating whether the sex of the evaluators affect a number of physical traits in handball players when they are tested in the presence of other players. More studies are definitely needed to explore the effect of the sex of the evaluators on physical performance in other conditions not explored in the present study.

COMMENT

On lines 268-270, you write, “Given that in the present study only 1 out of 15 correlations reached statistical significance, it is likely that the significant correlation may be anecdotal and promoted type I error.” What statistical adjustments did you use to correct for multiple comparisons? If you are concerned that Type 1 errors are a threat to the validity of your conclusions, what is the rationale for running these analyses? 

RESPONSE

We reported the r coefficients of all combinations in Table 2 because it does not seem reasonable to exclude any of the physical traits in these analyses. We highlighted in the table the only correlation that reached statistical significance (without any statistical correction). In the discussion we provide an interpretation of the results presented in Table 2. We believe that the existence of direct and indirect relationships (i.e., positive and negative r coefficients) and that only one of the r coefficients reached statistical significance, suggest that the sex of the evaluator did not have a consistent effect on individual subjects for the performance in the different physical tests. We believe that this information is important and that the results are properly interpreted. Please, note also that there is no consensus in the scientific literature regarding if the p-values of should be adjusted when multiple associations are computed in the same study. For that reason, we decided to highlight the only r coefficient that was significant without any correction and later we interpreted this finding in the discussion section.

COMMENT

7. Why were other members of the handball team present during testing? It seems like the study may be confounded by this. Though it is highlighted as a limitation, additional explanation for this decision should be provided. 

RESPONSE

Yes, it could be a confounding variable. However, it should be noted that team sport athletes are commonly tested in the presence of other members of the team. Therefore, we were interested in exploring the effect of the sex of the evaluator in the conditions that handball players are commonly assessed. 

COMMENT

8. Given the lack of control group and the presence of the handball team during exercise testing, I do not think that the following claim can be made: “The results of the present study collectively suggest that the sex of the evaluator is not an important factor to consider when assessing strength performance in young male athletes. Therefore, male handball players can be indistinctly assessed by male or female evaluators.” Though your study suggests that the majority of physical outcomes were not differentially impacted by male or female evaluation, more research is needed before we can confidently conclude that male handball players’ performances may not be significantly affected by the sex of evaluators. 

RESPONSE

We agree with the reviewer´s comment. We have modified the conclusion and now it reads: “The results of the present study collectively suggest that the sex of the evaluator is not an important factor to consider when the strength performance of young handball players is tested in the presence of other members of the team. Therefore, when physical tests are implemented in the presence of other members of the team, which is the most common procedure in practice, male handball players can be indistinctly assessed by male or female evaluators. Future studies should be implemented to explore the effect of the sex of the evaluators when athletes are tested without the presence of observers (e.g., other team members).” The conclusion has been also modified in the abstract.

COMMENT

On line 77, I think “adversary” should be changed to adversaries 

RESPONSE

Thank you. This change has been made. 

COMMENT

On line 288, I think “presented” should be changed to present 

RESPONSE

Thank you. This change has been made.

---

## [Decision Letter · Decision Letter 1]

29 Mar 2021

Ballistic, maximal strength and strength-endurance performance of male handball players: are they affected by the evaluator's sex?

PONE-D-20-34628R1

Dear Dr. Janicijevic,

We’re pleased to inform you that your manuscript has been judged scientifically suitable for publication and will be formally accepted for publication once it meets all outstanding technical requirements.

Kind regards,

Jeremy P Loenneke

Academic Editor

PLOS ONE

Additional Editor Comments (optional):

Reviewers' comments:

Reviewer's Responses to Questions

**Comments to the Author**

1. If the authors have adequately addressed your comments raised in a previous round of review and you feel that this manuscript is now acceptable for publication, you may indicate that here to bypass the “Comments to the Author” section, enter your conflict of interest statement in the “Confidential to Editor” section, and submit your "Accept" recommendation.

Reviewer #1: All comments have been addressed

Reviewer #2: All comments have been addressed

2. Is the manuscript technically sound, and do the data support the conclusions?

Reviewer #1: Yes

Reviewer #2: Yes

3. Has the statistical analysis been performed appropriately and rigorously? 

Reviewer #1: Yes

Reviewer #2: Yes

4. Have the authors made all data underlying the findings in their manuscript fully available?

Reviewer #1: Yes

Reviewer #2: Yes

5. Is the manuscript presented in an intelligible fashion and written in standard English?

Reviewer #1: Yes

Reviewer #2: Yes

6. Review Comments to the Author

Reviewer #1: In general, I think the authors did a very nice job addressing all of the reviewer comments. While I am still a bit concerned over the generalizability of the findings based on the specific male and female test administrators chosen, the authors have made these concerns clear in the limitations section and have added a substantial amount of information regarding the demographics of these test administrators. As such, I am okay with the manuscript given the transparency of the potential limitations.

Reviewer #2: The reviewer would like to thank the authors for addressing all previous comments. The manuscript has been improved greatly and will be of interest for many.

7. PLOS authors have the option to publish the peer review history of their article (what does this mean?). If published, this will include your full peer review and any attached files.

Reviewer #1: No

Reviewer #2: No

---

## [Editor Report · Acceptance letter]

13 Apr 2021

PONE-D-20-34628R1 

Ballistic, maximal strength and strength-endurance performance of male handball players: are they affected by the evaluator's sex? 

Dear Dr. Janicijevic:

I'm pleased to inform you that your manuscript has been deemed suitable for publication in PLOS ONE. Congratulations! Your manuscript is now with our production department. 

Kind regards, 

on behalf of

Dr. Jeremy P Loenneke 

Academic Editor

PLOS ONE